# Intense rains in Israel associated with the train effect

**Baruch Ziv[1], Uri Dayan[2], Lidiya Shendrik[1], and Elyakom Vadislavsky[3]**

[1]Department of Natural Sciences, The Open University of Israel, Raanana, Israel
[2]Department of Geography, The Hebrew University of Jerusalem, Jerusalem, 9070227, Israel
[3]Israel Meteorological Service (IMS), Beit Dagan, Israel

**Correspondence:** Baruch Ziv (zivbaruchana@gmail.com)

**Abstract.** The "train effect" is defined as a cloud system in which several convective cells pass over the same place in a short time. Trains produce a large amount of rainfall, frequently leading to flash floods, reported mainly over North America during spring and summer. Thirty train events were identified using radar images calibrated by rain gauges for four winters, all associated with Cyprus lows (CLs). The dynamic factors responsible for their formation in Israel were examined, utilizing the ECMWF Integrated Forecast System with 0.1° resolution.

Seventeen out of the 30 events share common features. Each one was found within the cold sector in the southern periphery of a CL at its occluded stage and located in the left flank of a maximum wind belt, where cyclonic shear vorticity exists. The trains cross the Israeli coast near 32.2° N, with a mean length of 35 km; last 2–3 h; and yield a total of 30–50 mm of rainfall. The maximum wind belts to the right of the trains were found to delineate the limit of the precipitative region of the CLs. Unlike classical trains, activated by thermal or frontal forcing, the eastern Mediterranean trains that develop in a cold air mass can be referred to as "cold trains" rather than the classical "warm trains".

## 1 Introduction

The Mediterranean lows that reach its eastern part are called Cyprus lows (CLs). These are mid-latitude cyclones (HMSO, 1962; Ulbrich et al., 2012) that are responsible for about 90 % of the annual rainfall in Israel (Goldreich et al., 2004). The daily rainfall associated with CLs is on the order of 10–30 mm (Striem, 1981; Saaroni et al., 2010), and the extreme values exceed 50 mm (Katsnelson, 1964; Sandler et al., 2024), sometimes within a couple of hours (Morin et al., 2007; Dayan et al., 2021). Hereafter, "rainfall" refers to the total amount (mm) over a certain period and time, and "rain rate" is expressed in units of $\mathrm{mm\,h^{-1}}$.

The cell propagation vector, also called the "train effect", refers to intense rains, "where cells form and pass repeatedly, in succession, over the same location, results from a linear organization" (Doswell III et al., 1996). The cloud elements composing the typical train belong to the family of mesoscale convective complexes (MCCs). The train is a mesoscale phenomenon, supported by synoptic-scale processes, mostly through "moistening and destabilization created by the modest but persistent synoptic-scale vertical ascent ahead of shortwave troughs" (Doswell III, 1987). Actually, the synoptic-scale and the mesoscale factors interact synergistically to form the lower-level convergence accompanied by upper-level divergence conditions that last for several hours (Chappell, 1986). Intensive mesoscale uplift alone is insufficient to activate the train without the support of the moderate uplift imparted by the synoptic factor. The mesoscale factor may be indiscernible but can be deduced from the inability of the synoptic ingredients alone to explain heavy precipitation associated with a train (Doswell III et al., 1996).

The train effect is associated with quasi-stationary cold fronts or within pre-frontal warm tongues, mainly during the spring and summer (Chappell, 1986; Doswell III et al., 1996). It has been reported mostly over North America (Schwartz et al., 1990; Corfidi, 2003; Wang and Chen, 2009). An essential factor for maintaining a train is a continuing transport of moist and unstable air. An effective system that supplies such a transport is the low-level jet (LLJ) that tends to form near cold fronts (Wang and Chen, 2009).

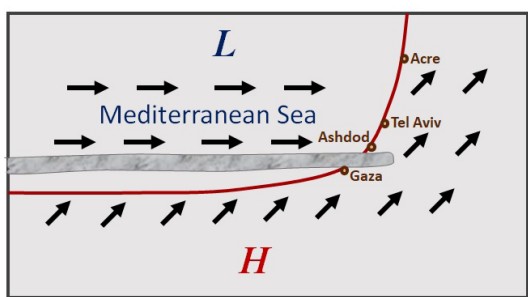

**Figure 1.** A schematic illustration of cloud strips producing rain over the southern coast of Israel. "L" and "H" denote cyclone and anticyclone, respectively; the arrows represent wind vectors; and the shading represents the cloud strip (following Fig. 6 of Rosenfeld and Nirel, 1996).

The train effect was also identified in western Europe. Rigo and Llasat (2005) analyzed a "convective train" associated with a mesoscale cyclone that produced a summer severe flash flood in southeastern Spain. ZAMG (2014) analyzed 100 convective systems over the Iberian Peninsula and France during the period 1992–2009, 7 of them in the form of trains, with lengths of hundreds of kilometers, all ahead of a cold front (i.e., pre-frontal train). One of them lasted for 16 h. As the major conditions for train generation, they mentioned LLJ near 925 hPa and convergence of specific humidity at 1000 hPa.

Heavy rains associated with cloud strips resembling the train effect have been identified in the eastern Mediterranean (EM). The cloud elements composing these systems were not in the form of MCCs but rather in the form of smaller or less persistent cells. These cloud systems were analyzed first by Rosenfeld and Nirel (1996), who considered them "coastal fronts" (e.g., Bosart, 1975), which are formed by convergence of land breezes. Rosenfeld and Nirel (1996) attributed these systems to convergence between a southerly land breeze originating from the North African and north Sinai desert coasts and the westerlies associated with a CL over the adjacent Mediterranean (Fig. 1). When such a cloud strip crosses the Israeli coastline, it could generate continuous rain. Goldreich et al. (2004) stated that the rain produced by this type of coastal front lasts 20 h on average and that the coastal fronts are most frequent in December, when the sea–land temperature contrast is the largest, and most active during the night and early morning hours.

The first time the term "train" was attributed to a rain system in Israel was by Dayan et al. (2021). They analyzed a severe flash flood that was generated by a series of consecutive convective rain cells within less than 2 h in southern Israel, within the cold sector of a CL in April 2018. The flash flood took place at Tzafit River in southern Israel (31.0° N, 35.3° E) and took the lives of 10 people. This case demonstrates the fatal potential of such a phenomenon for this region. It is worth noting that this train and those analyzed in the present study were found within the cold sector of mid-latitude cyclones, unlike the trains analyzed in North America and Europe, which were found within warm air masses and cannot be explained by the same mechanisms. We refer to them as "cold trains", in contrast to the "warm trains" previously studied.

The aim of this research is to identify and document trains that are associated with CLs and to analyze the major dynamic factors responsible for their formation. Section 2 specifies the data and methods used. Section 3 demonstrates the phenomenon in a case study and attempts to generalize its characteristics through composite maps. In Sect. 4, the results are discussed and summarized in a framework of a conceptual model in Sect. 5.

## 2 Data and methods

The study relies on four consecutive winters (December–February) of the years 2018–2022 (except January–February 2019 due to missing radar data) – a total of 10 months when CLs are most frequent (Alpert et al., 2004). The rain data are based on the radar operated by the Israel Meteorological Service (IMS) and calibrated by rain gauges using the INCA system integration method (Haiden et al., 2011).

The synoptic background and the mesoscale features for the cases analyzed are based on the atmospheric fields of the gridded data of the European Centre for Medium-Range Weather Forecasts (ECMWF) Integrated Forecast System (IFS) with 0.1° resolution (Hólm et al., 2016). This includes the sea-level pressure (SLP) and three wind components from all levels between 925 and 200 hPa. This data set was found to be superior to the ERA5 in resolving the signature of the trains in the rain field as well as in the vertical velocity field.

The train events were identified through inspection of the radar images for the rainy days included in the study period, in which CLs dominated the Levant region. An example is shown in Fig. 2, and the location of the CL is denoted in Figs. 1 and 2b. We used images with 5 min increments to search for repetitive passages of cloud cells producing an instantaneous rain intensity of $> 30\,\mathrm{mm\,h^{-1}}$ for 1 h or more. These are referred to as "train events" or "events" hereafter. In such a way, 30 events were identified, and for each of them an integrated rainfall map was extracted.

Compositing of atmospheric fields enables the combination of information from a number of examples of a phenomenon in a convenient format that highlights basic common features while eliminating details of individual events (Sinclair and Revell, 2000). The individual maps were transformed to bring them to a common basis so that in each one the cyclone center would be north of the train center and 500 km apart. This transformation includes rotation, translation and change of scale. The first step was to rotate each map about the location of the cyclone's center (which served as a

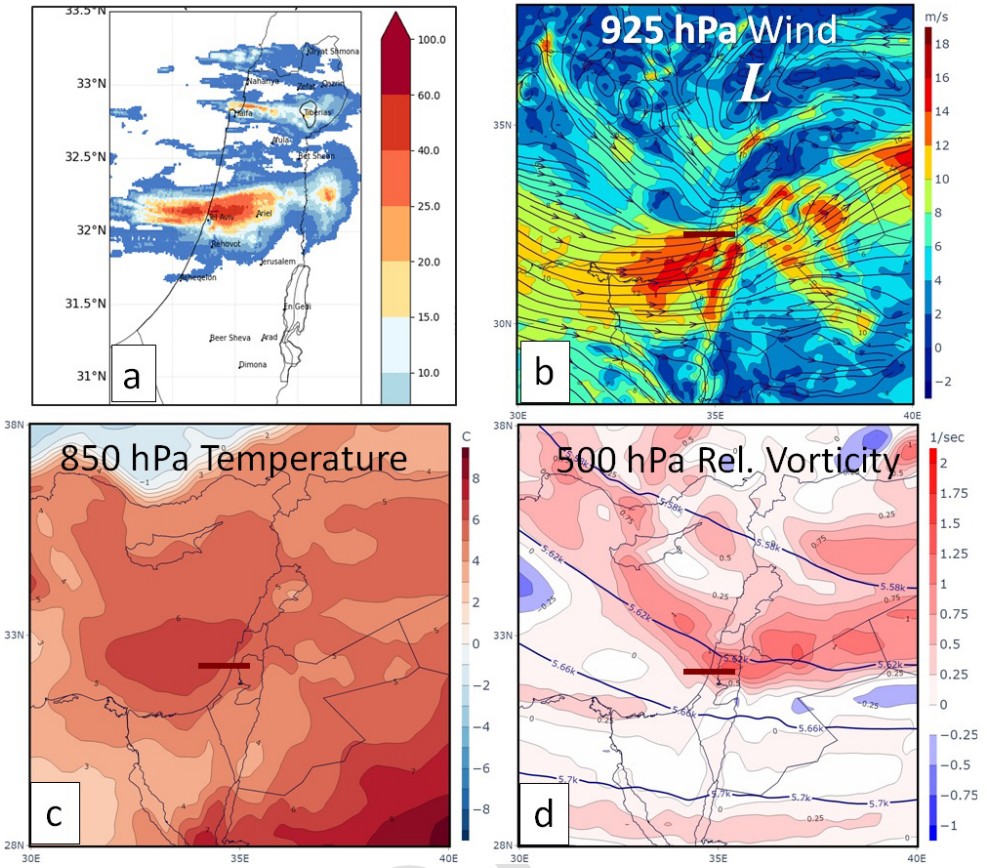

**Figure 2.** Synoptic conditions on 7 December 2018 of **(a)** integrated rainfall (Israel Meteorological Service) during 19:20–23:00 UTC. Atmospheric fields are from 21:00 UTC. **(b)** Wind field (streamlines and speed, in m s$^{-1}$ units) at 925 hPa. **(c)** Temperature (°C) at 850 hPa. **(d)** Geopotential height (GPH, m) and relative vorticity ($10^{-5}$ s$^{-1}$) at 500 hPa. The maps are derived from the ECMWF Integrated Forecast System (IFS) with 0.1° resolution. The train is denoted by a thick red line. Publisher's remark: please note that the above figure contains disputed territories.

reference point) by an angle equal (and opposite) to the deviation of the orientation of the line connecting the centers to the train and the cyclone from the south–north direction. The second step was to rotate the wind direction accordingly in every grid point. The third was to stretch (or contract) the grid to bring the distance between the centers of the train and the cyclone to 500 km. The locations of the CLs' centers were identified as the minimum at the 925 hPa geopotential height (GPH) closest to Cyprus. If such a minimum was not found, the nearest cyclonic vortex at that level was regarded as the CL center. The details of the variables for which composite maps were derived and the specific transformations used are specified in Sect. 3.

# 3 Results

In attempting to identify the mechanisms responsible for train events, we isolated from the 30 train events 17 that had a common feature. All of them were found to delineate the southern boundary of the rainy sector within the cold sector of CLs, in the vicinity of the band of maximum wind. These 17 trains were separated from the other 13 trains, which were not attributable to any other common one, and constitute the study sample. First, they are exemplified by a case study, and then they are analyzed collectively by composite maps.

## 3.1 A case study

During 6–8 December 2018 the EM was dominated by a CL (Fig. 2b and d), accompanied by an upper-level trough, whose axis was located to the east of the Levant (Fig. 2d) TS1. Rains were spread over the northern half of Israel, the region characterized by the Mediterranean climate type denoted "Cs" according to Köppen classification, i.e., "the thermal zones of the earth according to the duration of hot, moderate, and cold periods and to the impact of heat on the organic world" (Volken and Brönnimann, 2011). The rainiest area was the coastal plain, with 200–270 mm of rainfall, and the Judaean Mountains extending east of it (including

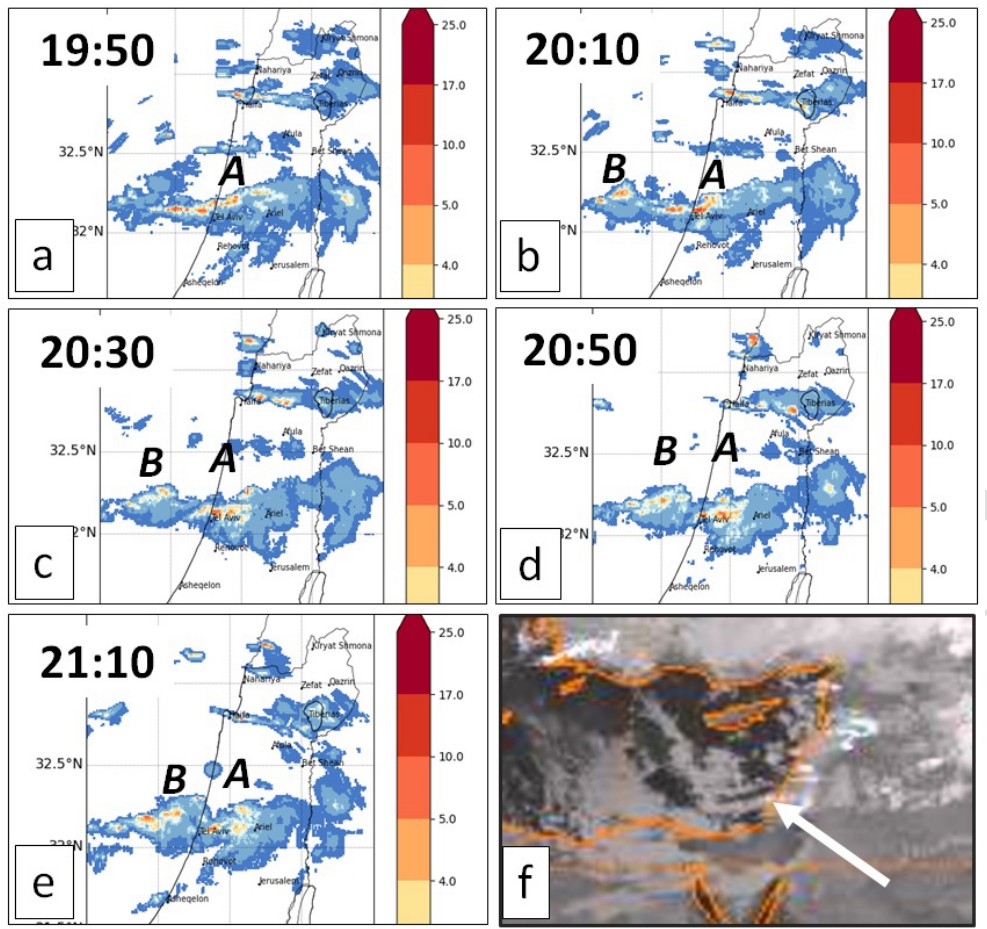

**Figure 3.** Cloudiness for 7 December 2018 **(a–e)**. Radar imageries (Israel Meteorological Service) at 19:50, 20:10, 20:30, 20:50 and 21:10 UTC, respectively, transformed to rain intensity (mm 10 min$^{-1}$); "A" and "B" refer to the first and second rain cells, both advancing eastward in tandem. **(f)** Meteosat 8 IR imagery for 21:12 UTC. The white arrow points at the train location. Publisher's remark: please note that the above figure contains disputed territories.

Jerusalem) that received half of this amount (IMS publication, 2018).

On 7 December the center of the CL was located at the northeastern corner of the Mediterranean, as can be inferred by a vortex at the 925 hPa level there (Fig. 2b). The rainfall during that day over the northern half of Israel was 30–40 mm, with a few peaks of $\sim 60$ mm over the coastal plain. On that day, at 19:30 UTC, a cloud strip in the form of a train, oriented in a west–east direction, developed across the central part of the coastline (Tel Aviv, 32.2° N, Fig. 3). This system persisted for 2.5 h and generated heavy rain, with a maximum instantaneous rate of over 60 mm h$^{-1}$ (as inferred from Fig. 3a–e), summing up to $> 50$ mm of rainfall during the entire event (Fig. 2a). Another train occurred over the region in the previous day (Table 1, no. 1) and yielded an average rainfall of 50 mm (not shown). This implies that the two trains contributed $\sim 35$ % of the total rainfall obtained over the coastal plain during the 3 d of the rainstorm. The train was 100 km long, $\sim 5$ km wide at its western end and $> 10$ km wide at its eastern end (Fig. 3). It was composed of three cells on the order of $< 10$ km; two of them are shown in Fig. 3a–e (denoted by "A" and "B").

The lower-level (925 hPa) wind field (Fig. 2b) shows that the EM and the Levant were under westerly flow, with cyclonic curvature and speed on the order of 5–10 m s$^{-1}$, except for a belt of $\sim 15$ m s$^{-1}$ over the southeastern Mediterranean. Confluence of the streamlines to the left of this belt is co-located with the train. The wind deviated 30° to the left of both the train orientation (Fig. 2b) and the tracks of the cloud cells (Fig. 3a–e). On the other hand, the isohypses at the 500 hPa (Fig. 2d) indicate that at that level the wind deviated 30° to the right of the train. This suggests that the clouds were stirred by the wind at the mid-troposphere. The lower-level temperature field (850 hPa, Fig. 2c) does not reflect any frontal structure and indicates that this train was developed within a homogeneous air mass to the south of the CL center. This and the absence of upper-level vorticity advection (Fig. 2d) indicate that the CL was at its occluded phase.

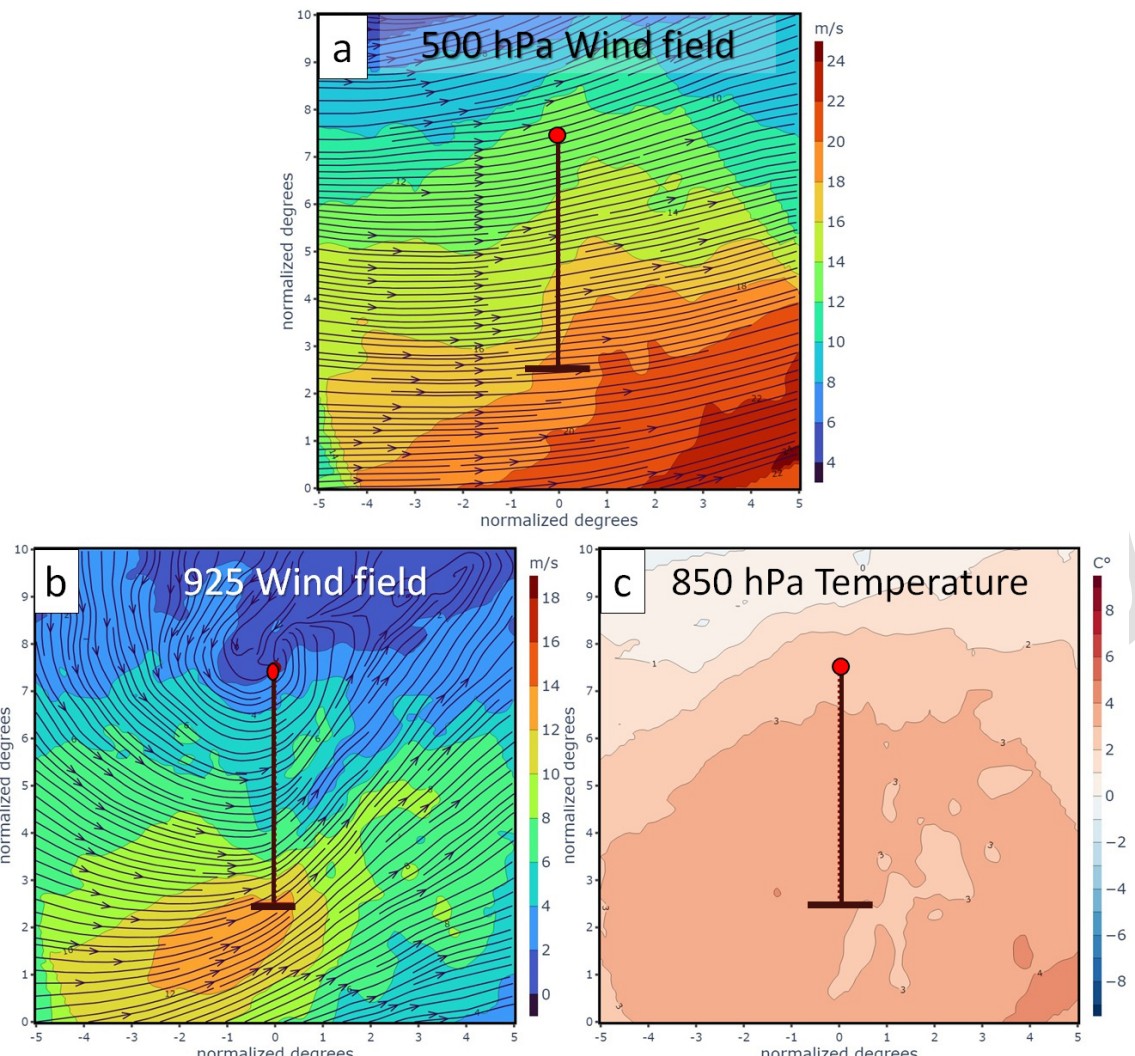

**Figure 4.** Composite map of the 17 members of the study sample. **(a, b)** Wind field (streamlines and coloring according to wind speed in m s$^{-1}$) at 500 and 925 hPa, respectively. **(c)** Temperature (°C) at 850 hPa. All individual maps are rotated, translated, and rescaled to bring the cyclone center and the train to the same location. In each map the train is denoted by a horizontal red bar and the cyclone by a small red circle, with a thick line joining them.

## 3.2 Common characteristics of the trains

The common features of the 17 trains included in the study sample are demonstrated through composite maps and specified in Table 1. The train events lasted between 1.3 TS2 and 5.1 h, with an average of 2.3 h, with no preferred time of the day for their occurrence. The average length of the trains was 44 km; five of them were > 60 km (maximum 100 km), and eight were < 30 km. The mean latitude where the trains crossed the coastal region of Israel is near 32.2° N, with a tendency to be concentrated on its southern two-thirds. Their centers were located at the coastline, within 0.5° around 34.8° E, suggesting that the coast plays a role in their evolution. A significant relation was found between the orientation and the latitude of the trains. The trains affecting the northern part of the coastline were oriented around 240–060°, while at the southern part their orientation was around 285–105°. A positive relation of 0.41, confident at the 0.10 level, was also found between the longitude of the cyclone center and the train orientation; i.e., trains associated with CLs east of the Mediterranean coast tended to be oriented northwest–southeast, while those associated with CLs to the west were oriented southwest–northeast. The maximum rainfall was between 20 and 70 mm, with an average of 35.5 mm. In 5 out of the 17 cases the values exceeded 50 mm, an amount which implies a high potential for flash floods in urban regions (e.g., Young et al., 2021). A substantial difference in precipitation activity was noted between the two sides of the train. In 13 of the 17 events, no rain was observed to the right (south) of

**Table 1.** Selected features of the trains included in the study sample. "Cyc" stands for cyclone, "lat" stands for latitude and "long" stands for longitude.

| No. | Date (dd/mm/yyyy) | Starting hour (UTC) | Duration (h) | Lat of train (° N) | Length (km) | Orientation (°) | Cyc center long (° E) | Cyc center lat (° N) | Maximum rainfall (mm)* |
|---|---|---|---|---|---|---|---|---|---|
| 1 | 07/12/2018 | 19:20 | 2.7 | 32.2 | 100 | 265 | 35.7 | 35.5 | 60 |
| 2 | 08/12/2018 | 03:00 | 2.5 | 31.7 | 20 | 270 | 35.4 | 36.2 | 60 |
| 3 | 05/12/2019 | 11:10 | 1.3 | 32.4 | 30 | 285 | 37.5 | 36.7 | 35 |
| 4 | 13/12/2019 | 13:30 | 2.0 | 32.9 | 30 | 240 | 36 | 36 | 45 |
| 5 | 13/12/2019 | 15:40 | 1.3 | 32.8 | 25 | 240 | 36 | 36 | 50 |
| 6 | 04/01/2020 | 02:00 | 2.5 | 32.6 | 65 | 240 | 34.1 | 34.5 | 45 |
| 7 | 04/01/2020 | 08:10 | 2.2 | 32.1 | 15 | 275 | 35 | 35.8 | 70 |
| 8 | 04/01/2020 | 11:30 | 1.0 | 31.7 | 20 | 280 | 35.1 | 36.3 | 35 |
| 9 | 05/01/2020 | 07:30 | 1.7 | 32.6 | 30 | 270 | 33.8 | 36 | 35 |
| 10 | 05/01/2020 | 11:00 | 4.0 | 32.6 | 100 | 270 | 35.8 | 35.5 | 55 |
| 11 | 08/01/2020 | 22:50 | 2.7 | 31.7 | 45 | 285 | 35 | 34.6 | 35 |
| 12 | 21/01/2020 | 19:10 | 2.2 | 32.0 | 20 | 350 | 36.5 | 33.5 | 35 |
| 13 | 18/12/2021 | 11:10 | 5.1 | 32.9 | 100 | 240 | 31 | 35.5 | 60 |
| 14 | 20/12/2021 | 19:40 | 2.3 | 31.9 | 85 | 285 | 35.5 | 36 | 28 |
| 16 | 16/01/2022 | 02:40 | 1.5 | 31.8 | 15 | 270 | 36 | 33.7 | 20 |
| 15 | 16/01/2022 | 05:50 | 2.2 | 31.7 | 20 | 270 | 36 | 33.7 | 45 |
| 17 | 26/01/2022 | 21:40 | 2.0 | 31.7 | 25 | 270 | 36.5 | 36 | 35 |
| | Average | | $2.3 \pm 1.0$ | $32.2 \pm 0.5$ | $44 \pm 32.4$ | $271 \pm 26$ | $35.3 \pm 1.4$ | $35.4 \pm 1.0$ | $35.5 \pm 13.5$ |

* The maximum rate (calculated in 5 min intervals) with respect to the duration of the event over the affected area.

the train. Also, in the remaining four events, the rain activity south of the train was much weaker than to the north of it.

Figure 4 shows the synoptic configuration under which the trains developed. The 500 hPa wind field (Fig. 4a) indicates the presence of an upper-level trough, slightly west of the CL, with a nearly westerly wind direction. This direction corresponds to that of the train and the movement of the individual cloud cells composing it (see Sect. 3.1). The lower-level wind field (Fig. 4b) demonstrates the criterion according to which the study sample was selected, i.e., a strip of maximum wind just to the right of the train. At the medium and higher levels, as exemplified by the 500 hPa wind field (Fig. 4a), such a linkage cannot be noted. This implies that the mechanism responsible for the train formation is found within the lower levels; see discussion below. The 850 hPa temperature (Fig. 4c) shows a homogeneous temperature over the EM, around 3 °C, which is below the long-term mean for each of the months included in the study period. It should be noted that similar homogeneity was seen in each of 16 out of the 17 cases, as exemplified in Fig. 2c TS3. This eliminates the possibility that the trains were activated by fronts or warm air masses (see Sect. 1).

Figure 5 presents the wind field with which the trains are associated. Figure 5a emphasizes the location of the train to the left of the maximum wind belt (see also Fig. 4b), where cyclonic shear vorticity exists. The positive vorticity to the left of this belt (elongated red band in Fig. 5b) stands in contrast to the negligible vorticity to its right. The absence of negative relative vorticity there can be explained by a cancellation of the negative shear vorticity by a positive curva-

ture vorticity imparted by the CL all over the region. The vertical–meridional cross section through the western part of the train, where the cloud cells evolve (Fig. 5c), shows a distinct ascending current, with a maximum of 10–20 cm s$^{-1}$ (rather large in mesoscale terms) between 800 and 700 hPa levels. The linkage between this updraft and the lower-level positive vorticity is elaborated in the following section.

## 4 Dynamical considerations

The analysis of 17 rain systems identified in the Levant, which meet the definition of the train effect, enabled us to characterize their main features. A dynamical framework for the cold train is proposed as follows: it develops along the left flank of a maximum wind belt, where cyclonic shear vorticity exists along the southern periphery of a CL (Figs. 2b, 4b and 5a). They can be explained by an updraft (Fig. 5c) generated by a combination of cyclonic wind shear at the lower level and surface friction. The mechanism is demonstrated schematically by Fig. 7. If no friction exists and the region is subjected to a south-to-north pressure drop, the pressure gradient, which is purely meridional (Fig. 7a), produces nondivergent geostrophic westerly winds. The magnitude of the pressure gradient attains a maximum in the middle of the domain, yielding a belt of maximum wind across the domain.

When friction is imposed under the same pressure field, the wind weakens and deviates to the left from its westerly direction (Fig. 7b). As a result, the weaker winds to the right of the maximum wind band blow toward the faster winds

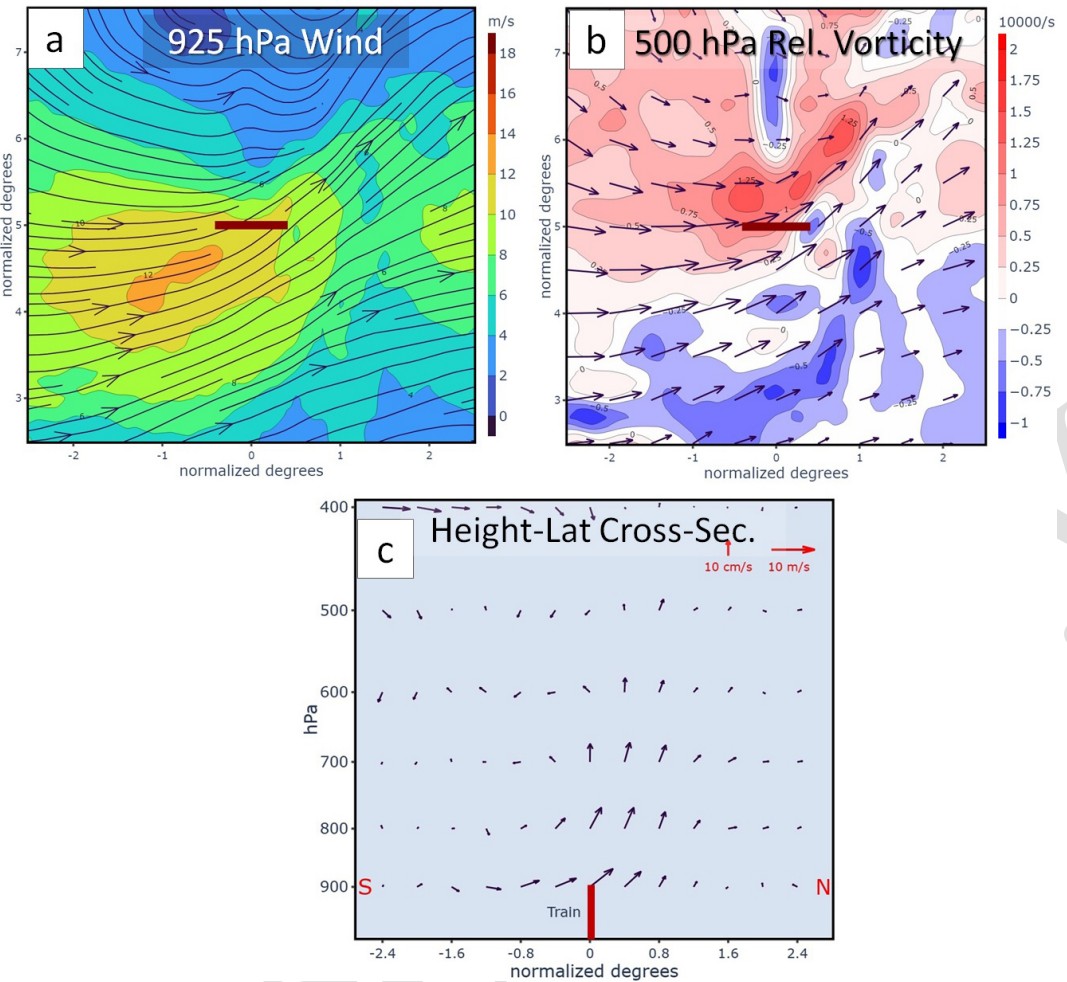

**Figure 5. (a–b)** Composite maps but based on rotation and translation only, so as to bring the train to a zonal orientation. **(a)** Wind speed and direction at 925 hPa. **(b)** Wind and relative vorticity ($s^{-1}$) at 500 hPa. The coordinates are degrees relative to train center. **(c)** Vertical–meridional cross section 10 km west of the trains' centers of the meridional and vertical wind components. The $x$ coordinates denote the latitude relative to the train, denoted by a thick red segment. The $y$ coordinate is pressure, in hPa.

along the core of maximum wind, and the fast winds at the maximum wind belt blow toward the weaker winds to the left of the core. The result is a band of along-stream convergence to the left of the core of maximum wind and divergence to its right. The elongated convergence band to the left of the maximum wind provides the uplift required for train formation.

The co-existence of lower-level positive vorticity, the upward motion and the resulting train indicate that friction is effective over the sea surface where the trains are created. It should be noted that in most of the events the core of the maximum wind belt delineated the southern boundary of the active sector of the CL, in terms of cloudiness and precipitation. This type of train reflects an interlace between the synoptic scale (the CL) and the mesoscale (the maximum wind belt), in accordance with Chappell (1986), Doswell III (1987) and Doswell III et al. (1996). Both contribute to the updraft

that produces this rain system, and the mesoscale factor determines its location and orientation.

The friction exerted by the sea surface seems questionable. Indeed, during calm conditions smooth sea surface produces negligible friction. But under the influence of a CL, with its induced winds, waves are created, which may enhance friction. Wave height measurements at the Israeli coast on the days belonging to the study sample were taken from the Coastal and Marine Engineering Research Institute Ltd (CAMERI) in Israel (https://www.cameri-eng.com/, last access: 1 June 2024). They show an average significant height of $2.41 \pm 0.61$ m, with an average maximum of $3.22 \pm 0.67$ m. This implies the sea state is rough, suggesting that friction cannot be ignored. Moreover, the presence of cumulus clouds associated with the CL creates turbulence that contributes friction as well. The combination of the factors that generate the trains is presented schematically in Fig. 8.

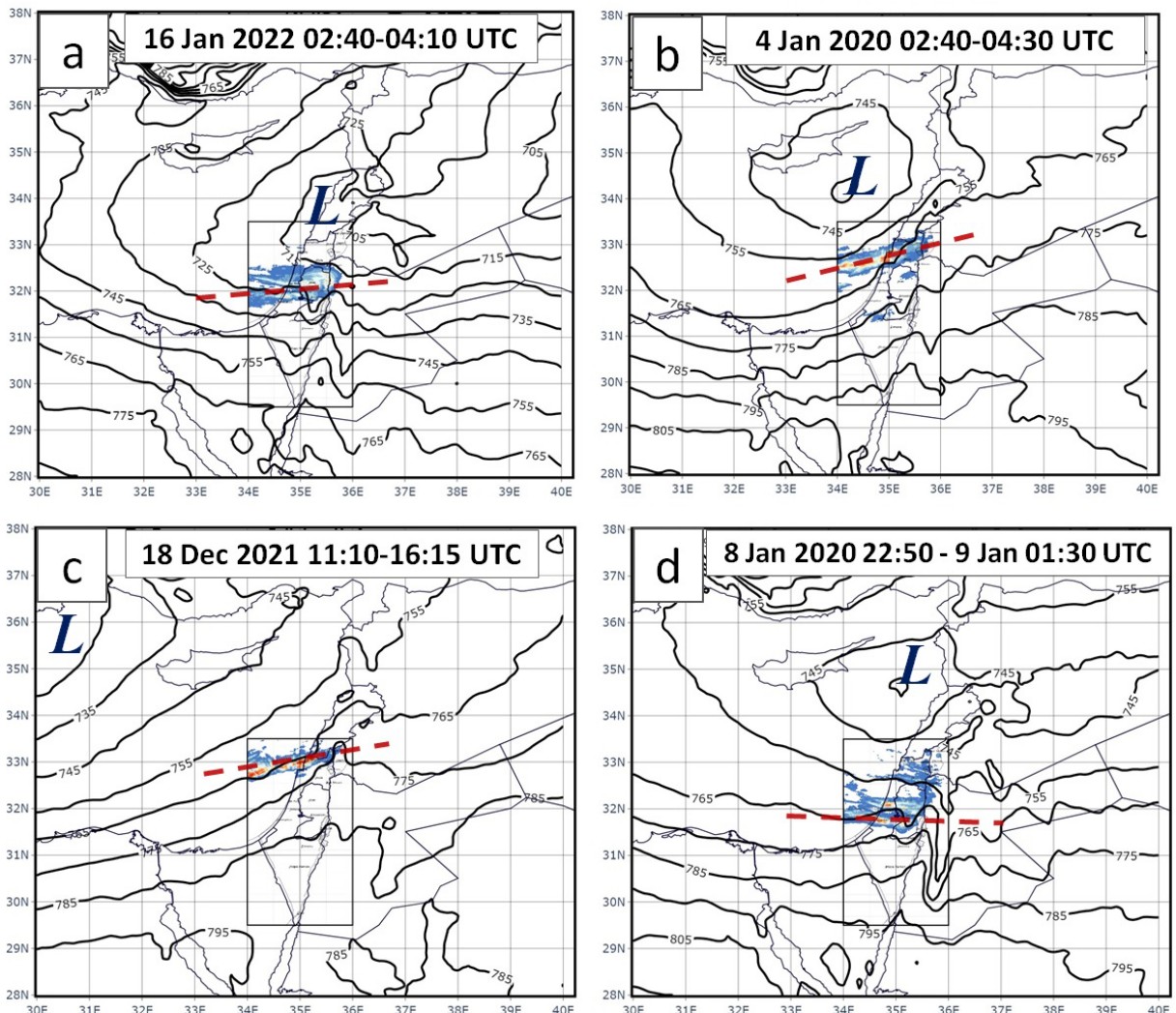

**Figure 6.** Integrated rainfall superimposed on geopotential height for **(a)** 16 January 2022, 02:40–04:10 UTC; **(b)** 4 January 2020, 02:40–04:30 UTC; **(c)** 18 December 2021, 11:10–16:15 UTC; and **(d)** 8 January 2020, 22:50 UTC, to 9 January 2020, 01:30 UTC. In each panel, the rectangles encircling the rain maps denote the radar coverage, the letter "L" denotes the cyclone center, and the dotted red line denotes the location and orientation of the train. Note that only in panel **(b)** there is some rain south of the train.

## 5   Summary and discussion

The trains associated with CLs, analyzed previously, are of mesoscale dimensions, i.e., $\sim 35$ km long and 10–20 km wide, persisting for 2–3 h, and yielding 30–50 mm of rainfall. They were referred to as cold trains since they were located within the cold sector of mid-latitude cyclones. They were found mainly in the winter, December–February, though such an event was documented in the spring (see Sect. 1).

The cold trains and the classical warm trains share a common feature, i.e., lower-level uplift, which initiates deep convection under preexisting static stability. However, they differ in several aspects. The major difference between them is in the identity of the factor initiating their formation. For the trains associated with fronts, the factor is buoyancy due to temperature difference across the front, whereas for the cold trains, it is forcing exerted by lower-level convergence. Another difference is in their seasonality. While the warm trains are formed in the spring and summer in North America (Doswell III et al., 1996) and in western Europe (e.g., ZAMG, 2014), the trains in the EM are frequent mainly in the winter, when CLs prevail. The two types of trains differ also in their dimensions. The typical classical warm trains are $\sim 10$ times longer, are 5 times larger and last several times longer than the cold trains. The comparison is summarized in Table 2.

The trains analyzed in this study share common features with the coastal fronts related to CLs, studied by Rosenfeld and Nirel (1996) and by Goldreich et al. (2004). Both are found over the southern part of the EM, zonally oriented (par-

**Table 2.** Comparison between the classical warm trains (see Sect. 1) and the trains analyzed here.

| Characteristic | Typical length (km) | Size of individual cells (km) | Timescale | Main season | Synoptic and thermal background | Major factor |
| --- | --- | --- | --- | --- | --- | --- |
| Classical warm train | Several hundreds | 50 | Half a day | Warm season | Quasi-stationary cold front, warm tongue | Spontaneous, via buoyancy |
| Cold trains analyzed here | Less than 100 | 10 | Several hours | Cold season | Cold air mass in occluded cyclone | Forced, to the left of the maximum wind belt, in the southern margins of a CL |

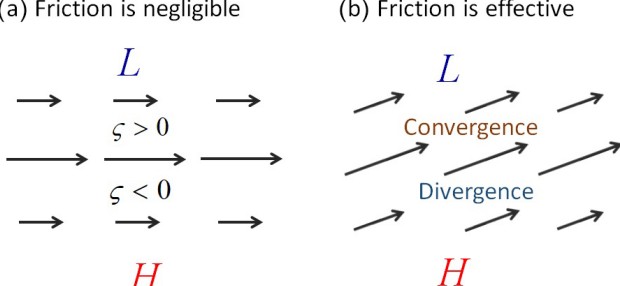

**Figure 7.** Schematic description of the wind pattern associated with a zonally uniform south-to-north pressure gradient producing a band of maximum wind: **(a)** when friction does not exist, **(b)** when friction is imposed, and when the wind speed is reduced by 30 % and its direction is deflected by 30°.

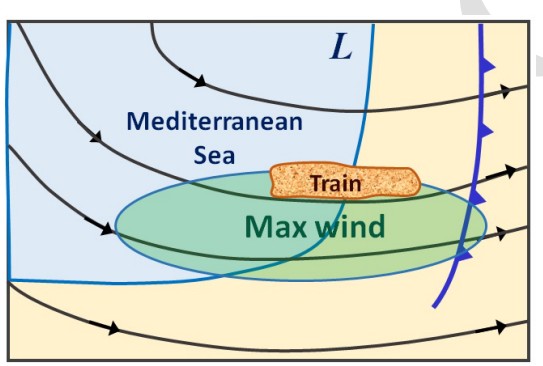

**Figure 8.** Schematic description of the lower-level configuration of the train associated with Cyprus lows. The background covers the eastern part of the Mediterranean, the Levant and the coastal region of the Sinai desert. The black lines represent isobars, and the arrowheads represent the direction of the geostrophic wind. The blue "L" denotes the CL center, and the thick line with teeth represents the cold front. The region of the maximum wind within the lower 1 km and the train itself are denoted.

allel to the North African coast). They cross the Israeli coastline and produce continuous, and sometimes heavy, rains, with total rainfall that in some cases exceeds 50 mm. How-

ever, several substantial differences can be noted between these two phenomena.

The term "coastal front" was coined by Bosart (1975) for cloud bands that develop over the sea, near and parallel to coastlines, as an outcome of land–sea contrast in temperature or friction. Rosenfeld and Nirel (1996) explained the formation of the coastal fronts as resulting from a convergence band where the warmer sea generates a land breeze from Egypt. This breeze blows toward the Mediterranean Sea and confluents offshore, with the nearly geostrophic westerly flow over the warmer sea surface (Fig. 1). They based their explanation on friction and temperature contrast. Our analysis did not reveal any lower-level-temperature land–sea contrast across the North African coast, as is reflected in the temperature map at 850 hPa (Fig. 4c) and at 925 hPa (not shown). Moreover, the occurrence of the trains does not show any diurnal maximum during the night and morning hours (Table 1), when the land breeze is most intense. The coastal fronts are located close to the North African coast and are oriented parallel to it (250–275°). Consequently, they intersect the EM coast within a narrow segment of ∼ 50 km, between Gaza and Ashdod (Fig. 1). In contrast, the trains' orientation is highly variable, between 240 and 350°, with a standard deviation of 26° (Table 1), and the EM coast segment affected by them is much wider – about 150 km. This implies that, unlike the coastal fronts, the trains do not owe their existence and location to the North African coast.

The coastal fronts differ from the trains also in their frequency and dimensions. The number of coastal fronts observed by Rosenfeld and Nirel (1996) is 6 in 11 winters, reflecting a small frequency compared to the 17 train events identified as CE1cold trains within less than four winters in this study. As for the dimensions, the coastal fronts are larger than the trains. Their horizontal dimensions are ∼ 3 times larger, and they last 9 times longer. It can be concluded that in spite of the apparent similarity between the coastal fronts and the trains, both associated with CLs, these are different phenomena.

This research sheds light on a specific source of flash floods: CL-induced trains, which hit the coastal region of Israel in the winter. This phenomenon gave us an opportunity

to inquire into the inner structure of the wind field associated with the CLs and its dynamic implications. It is suggested that the type of maximum wind belt with which these trains are associated is an ingredient of the CL and that in most of the events it denotes the southern limit of its precipitative region. The results of our study identify a type of region that is prone to continuous heavy rains through the wind field. The precision of the regional models in mapping the mesoscale structure of the wind field enables us to point to the left side of maximum wind bands south of CLs as potential regions for rain-inducing flash floods.

*Data availability.* Pressure level NWP fields in NetCDF format are based on the post-processed ECMWF IFS model (https://www.ecmwf.int/, last access: 12 July 2023, Hólm et al., 2016). Meteosat Second Generation IR imageries were retrieved from the EUMETSAT data center (https://navigator.eumetsat.int/product/EO:EUM:DAT:MSG:HRSEVIRI, Kerkmann, 2019).

*Author contributions.* BZ led the dynamic aspects of the study, developed its methodology and contributed to writing the manuscript. UD conceived the study, set the goals of the article and contributed to writing the manuscript. LS performed the data processing, performed the calculations required for derivation of the composite maps, and helped in the preparation of the figures. EV processed the integrated radar-based rain maps.

*Competing interests.* The contact author has declared that none of the authors has any competing interests.

ther geographical representation in this paper. While Copernicus Publications makes every effort to include appropriate place names, the final responsibility lies with the authors.

*Acknowledgements.* We would like to thank Yoav Levy, head of the research and development department of the Israel Meteorological Service, for the radar and integrated rain maps. The authors express their gratitude to CAMERI, the Israeli Coastal and Marine Engineering Research Institute Ltd (https://www.cameri-eng.com, last access: 1 June 2024), for contributing the wave data used in this research. We are grateful to the Israel Meteorological Service (IMS) for providing the IMS C-band Doppler radar imagery and the integrated rain maps.

*Review statement.* This paper was edited by Gregor C. Leckebusch and reviewed by two anonymous referees.

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

**Remarks from the language copy-editor**

CE1     Please confirm change.

**Remarks from the typesetter**

TS1     Please confirm.

TS2     Please note that an explanation is needed here why the values should be changed. Therefore, please provide a \*.pdf file with sticky notes which we can forward to the handling editor. Without the explanation and the approval of the editor, it is not possible to change the values. Thank you for your understanding.

TS3     Please confirm.

TS4     Please note that this URL has been taken from the data availability section. Please confirm that it is correct.