# Peer review of "Intense rains in Israel associated with the 'Train effect'"

_Natural Hazards and Earth System Sciences, 2023_

## Community Comment (CC1)

**REFEREE'S COMMENTS TO THE EDITOR AND AUTHORS**

Manuscript (nhess-2023-215) entitled: Intense rains in Israel associated with the 'Train effect', by Ziv et al.

**General**

This article presents a comprehensive investigation of rain events associated with the so-called "Train Effect" phenomenon, observed in the eastern Mediterranean. The authors analyze dynamical factors and phenomenological features of 17 events, all found in the southern periphery of occluded Mediterranean cyclones, using case study analysis and composite maps. The article presents two new findings and subsequent insights, for which it deserves to be published. One, the essential differences between "East-Mediterranean trains" and those studied in the US and Western Europe. The second, the difference characterized between "coastal front" events studied in the coastal area of Israel (by Rosenfeld and Nirel 1996) and the train phenomenon studied in the presented article.

The article establishes a theoretical framework for explaining "trains" as an inherent part of active cloudiness associated with occluded East Mediterranean Cyclones (i.e., Cyprus Lows). Therefore, I recommend publishing the article in Nat. Hazards and Earth System Sciences, after some weaknesses and mistakes are corrected, as specified below.

**Major Comments**

1. The study included 17 events out of 30 identified. It is essential to refer to the **13 events** that were not investigated and explain why they were not investigated. It must be explained whether they represent a different phenomenon or dynamic process, which may change the insights, or whether they were less prominent in their characteristics.

2. The authors note that based on previous studies: ""the train effect is associated with quasi-stationary cold fronts or within pre-frontal warm tongues…" (line 39). The dynamic factors and process leading to the formation of trains associated with **quasi-stationary cold fronts** in the cold front must be explained, this is to distinguish it from the different dynamical factors associated with Cyprus Lows.

**Specific comments:**

Page 2, lines 43-44: the authors mention "Train effect was identified also in Western Europe, near the western Mediterranean". Be more specific about the location.

Page 2, lines 45-46: "ZAMG (2014) analyzed 100 convective systems during the period 1992 – 2009…". Specify the location of this study.

Page 2, line 55: "from the North African coast…" change to: "from the North African and North Sinai Desert coasts…"

Page 2, line 57: "…the rain produced by this type of system lasts 20 hours…" For the sake of clarity specify which type of system he refers to.

Page 2, line 67: "…within the cold sector of a CL in 2018." Add **April** 2018 and stress that 'cold trains' occur not only in the mid-winter but also during spring.

Page 3, line 70: "…to document trains". I recommend changing to "… identify, document and analyze…"

Page 3, line 76: "…four consecutive winters (December-February) of the years 2018-2022 (except January-February 2019 due to missing radar data)." Change to: "…four consecutive **mid-**winters (December-February) of the years 2018-2022 (except January-February 2019 due to missing radar data), **total of 10 months**."

Page 3, lines 80-83: Why did the authors utilized data with high resolution, of 0.1°, and did not use the ERA5 data of 0.25° resolution?

Page 4, line 112: "characterized by Mediterranean climate". Please specify what are the characteristics of this type of climate and add a citation.

Page 4, line 112: "The rainiest area was the coastal plain, with 200-270 mm…".  State how long this amount of rain accumulated and state the ratio between the amount that fell as a product of the trains (see line 120) and the total rainfall depth.

Page 4, line 120: "…summing up to > 60 mm for the entire event". Were there other trains that were not identified or investigated? I recommend referring to the existence of parallel trains.

Page 5, lines 136-137: "…was developed within a homogeneous air-mass". Recommend changing to: "…was developed within a homogeneous **cold** air-mass".

Page 5, line 138: "…indicate that the CL was at its occluded phase."  I recommend adding a sentence (here or in the discussion section) that explains the difference from the quasi-stationary cold front, mentioned in the introduction section.

Page 7, lines 161-162: "In 13 of the 17 events, no rain was observed to the right (south) of the train." And in the other four events, was there a difference in precipitation activity?

Page 8, Table 1: titles: add °N to the Lat and °E to the Lon.

Page 8, Table 1: It is recommended to add the standard deviation in addition to displaying the average only.

Page 9, Fig. 4: After a thorough inspection I have noticed that the location of the train left of the maximum wind takes place only on the lower level (925-hPa level). Please, point to the fact that the mechanism responsible for the train originates from the below. Moreover, the figure will become clearer if you add notation of the pressure-level on each part of it.  Also, why is the temperature (or temperature anomaly) plot not shown at the 850-hPa level and the 500-hPa GPH and vorticity, as shown in Figure 2?

Page 10, Fig. 5: For the sake of clarity, I suggest adding notation of the pressure-level and atmospheric variable in parts a and b.

Page 11, line 234: "…show an average significant height of 3-4 meters, with an average maximum of 5-6 meters". A citation is needed.

Page 12, lines 264-265: "…a small frequency compared to the 17 train events..." Shouldn't it be 30 events?

**Typographical errors:**

Page 1, line 19: change EM trains to East Mediterranean trains

Page 1, line 27: Saaroni et al 2011 should be corrected to Saaroni et al. 2010 and Sandler et al. 2023 should be corrected to Sandler et al.2024.

Page 1, line 32: delete the sign ' after the word mostly.

Page 2, line 43: Change "tend" to "tends".

Page 2, line 45: Change "in the form of train" to "in the form of trains".

Page 2, line 46: "in the form of A train" (missing "a") or "in the form of TRAINS" (in plural).

Page 2, line 50: Change "train effect has been…" to "train effect have been".

Page 3, line 68: Change "took lives of 10 people" to "took the lives of 10 people".

Page 3, line 71: Change "attempt" to "attempts".

Page 3, line 77: "The rain data is based" to "The rain data are based".

Page 3, line 87: Change "These are regarded" to "These are regarded as".

Page 3, line 94: Change "modified as to fit" to " modified to fit".

Page 3, line 95: Change "location of the CLs' centers" to "locations of the CLs' centers".

Page 5, line 134: Change "the cloud were stirred" to "the clouds were stirred".

Page 6, line 149: Change "5 of them are" to "5 of them were".

Page 7, line 157: Change "while these associated" to "while those associated".

Page 7, line 159: Change "flashflood" to "flash flood".

Page 11, line 228: Change "that produce this" to "that produces this".

Page 12, line 240, caption of Fig. 6: add the word Desert after Sinai so it will be "Sinai Desert".

Page 12, line 256: "… as is reflected in 925 hPa" Should be changed to: "as is reflected in the 925-hPa temperature map".

Page 13, line 269: Cl should be corrected to CL.

Page 13, line 272: Change "ingredient the CL" to "ingredient of the CL".

Lines 35, 40, 229: Change Chappel to Chappell.

Lines 31, 38, 229: Doswell 1996 should be changed to Doswell et al. 1996.

Delete the comma after the author's name in the following citations: L. 31: Doswell et al. 1996; L. 35: Chappell 1986; L. 41: Corfidi 2003; L. 77: Alpert et al. 2004; L. 80: Haiden et al. 2011; L. 83: Hólm et al. 2016; L. 92: Sinclair and Revell 2000; L. 114: IMS publication 2018.

Page 14, line 316: delete the reference of Alpert et al. (1990) since it does not appear in the text.

Page 14, lines 323-324: The link to Chappell (1986) cannot be found and should be: https://link.springer.com/chapter/10.1007/978-1-935704-20-1_13

Page 15, line 364: The reference of Saaroni et al should be 2010 (and not 2011).

Page 15, line 365: The reference of Schwartz et al. (1990) should be moved after the reference of Sandler et al.

Page 15, line 368: The reference of Sandler et al., should be moved before Schwartz et al. (1990), and it needs to be corrected to: Sandler, D., Saaroni, H., Ziv, B., Hochman, A., Harnik, N. and Rostkier-Edelstein, D:  A multiscale approach to statistical downscaling of daily precipitation: Israel as a test case, Int. J. Climatol., 44(1), 59-71, 2024. https://doi.org/10.1002/joc.8315

---

## Author Response (AR1)

**Intense rains in Israel associated with the 'Train effect' (nhess-2023-215)**

**Point to Point response to the reviewers**

All corrections requested were carefully treated. The corrections done are specified in red.

**Reviewer #1**

**General comments:**

This paper summarizes the results of a well designed and well carried out study into the characteristic dynamical feature associated with a "Train Effect" phenomenon in the eastern Mediterranean during winter. The authors identify the main dynamical features associated with the a train of precipitating convective cells moving from sea to land in the southern coastal region of Israel and compare and contrast them to features identified in previously studied train effect in the U.S. summer.

The paper can benefit from editing to improve its presentation quality by attending to grammatical and sentence structure errors. I indicated some of those below.

**Specific comments (in order of appearance in the text) our responses – in red:**

1. Page 1, line 18: "yield ~35 mm rainfall". Here and in the paper text the authors refer to rain amounts and it is not clear if the numbers imply rain rate (i.e., about per unit time) of the amount of precipitation falling over the entire life time of the phenomenon. I suggest being more specific regarding the values given here and in other cases in the text.

   In the pertinent place we changed to "yield total of ~35 mm rainfall". In the paper itself, in the first time when appears, we note that 'rainfall' refers to total amount (mm) over a certain period and time, and 'rain rate' is expressed in mm/h units.

2. Page 1, lines 29-30: Where is the sentence part put in quotes taken from (citation is missing)?

   The source is Doswell 1996, who is quoted one sentence below. For the sake of clarity, we moved it to the end of this quotation.

3. Page 4, line 112: "characterized AS A Mediterranean-TYPE climate", can you provide reference for "Mediterranean-type Climate"?

   After "..climate" we added: "type, denoted 'Cs' according to Koeppen classification, i.e., "The thermal zones of the earth according to the duration of hot, moderate and cold periods and to the impact of heat on the organic world" (Volken and Brönnimann. 2011).

Volken, E.; Brönnimann, S Meteorologische, Zeitschrift (2011). 20 (3): 351–360. Bibcode:2011MetZe..20..351K. doi:10.1127/0941-2948/2011/105", also in the reference list.

4. Page 4, lines 119-121: "persisted 3.5 hours and generated heavy rain, with a maximum of > 60 mmh -1(as inferred from Fig. 3a-e), summing up to > 60 mm for the entire event (Fig. 3f). The integrated rainfall during the event exceeded 40 mm along the train." The numbers don't' make sense to me.

   Thank you for this comment. The corrected sentence is: "This system persisted 3.5 hours and generated heavy rain, with a maximum **instantaneous rate of over 60 mm h$^{-1}$** (as inferred from Fig. 3a-e), summing up to >50 mm rainfall for the entire event (Fig. 2a)."

5. Page 9, sentence on lines 184-187: This is confusing. In Fig 4A the train is south of the maximum wind, i.e., to the right of it. The situation is different in Fig. 4b, where the train is on the left. So the relation of the train to the maximum wind depends on the level of the wind, right?

   Thank you for your valuable comment. The train was found left of the maximum wind at the lower level (925 hPa, Fig.4b, original version), but not at 700 hPa. Following another reviewer, we replaced the 700 hPa level by 500 hPa wind map, which also does not reflect a clear relation between the train and a band of maximum wind. Following your comment we added the following to the text: "At the medium and higher-levels, as exemplified by the 500 hPa wind field (Fig. 4a), such a linkage can not be noted. This implies that the mechanism responsible for the train formation is found within the lower-levels, see discussion below."

   Page 11, lines 228-29: So there is a common dynamical element between the two phenomena?

   There is only little dynamic element in common between the 'cold' train studied here and the 'warm' trains previously studied, as can be inferred from Table 2. We added the following sentence: "However, they share a common feature, i.e., lower-level uplift, which initiates deep convection under preexisting static stability."

   **Editorial comments:**

Page 2, line 43: in reference to the LLJ - change "tend" to "tends".

Corrected.

Page 2, line 46: "in the form of **A** train" (missing "a") or "in the form of TRAINS" (in plural).

Corrected.

Page 2, line 50: "Heavy rains …….. **HAVE** been identified"

Corrected.

Page 2, line 53: "convergence of **A** land breeze." (Missing "a") or "convergence of land **BREEZES**." (Refer to the land grease phenomenon in plural.

Changed to: "land breezes".

Page 2, line 54: "convergence between **A** southerly land-breeze"

Corrected.

Page 3, line 68: "took **THE** lives of 10 people.

Corrected.

Page 3, line 73: "and **ATEMPS** to generalize"

Corrected.

Page 3, line 77: "The rain data **ARE**"

Corrected.

Page 3, line 77: "with 200-270 mm" - over what time interval? The entire time span of the phenomenon?

This phrase appears in line 112. The previous sentence starts with noting the period of the event, 6-8 Dec, meaning 3 days.

page 4, line 117: 60 mm over what time?

Following the previous sentence, at the beginning of the paragraph, referring to 7 December, we modified the pertinent sentence to: "The rainfall during that day over the northern half of Israel was 30-40 mm, with a few peaks of ~60 mm over the coastal plain."

Page 4, lines 117-118: "form of **A** train"

Corrected.

Page 4, line 122: "**ON** the order of" (same on page 5 line 131).

Corrected in both places.

Page 6, line 148: Use "included in the study" instead of "composing the study"

Modified accordingly.

Page 7, line 155: Use "orientation was around" instead of "orientation is around" (past tense fits better with the previous and following sentences).

Corrected in two locations.

Page 7, line 130: "which **IMPLIES** a high potential"

Corrected.

Page 10, line 211: Instead of using "the factors are irrelevant" ), i.e., as if there are factors like that here and they are not material, I suggest stating that: "The factors relevant for EM trains are different"

Thank you for this comment. We modified the phrase to "Different factors are relevant for the EM trains".

Page 11, line 220: Instead of using the terms "left flank" or "right flank" here and before why not use more geographical term such as "north flank" and "south flank"?

The terms 'left' and 'right' were chosen for stressing the general dynamic implications of the band of maximum wind, which are better described in terms of natural coordinates.

**Reviewer #2**

**Specific comments:**

Lines 81-84: Is there any reason this dataset is used instead of ERA5?

We had tried to use the ERA5 with 0.25° resolution, but found that it poorly reflects the signature of the trains. We added to the text the following sentence: "This data set was found superior over the ERA5 in resolving the signature of the trains in the rain field as well as in the vertical velocity field."

Lines 85-89: It might be useful to include a schematic diagram to better illustrate how this is defined in the current study.

For the reader convenience, and better clarity, we refer now the reader to Figs. 1 and 2b (new version), in which a Cyprus Low and a train are shown, as follows: "An example is shown in Fig 2, and the location of the CL is denoted in Fig. 2a,b".

Lines 92-94: I am a bit confused here. What is the "reference point" (e.g. is it "train"-centred?) of these transformations? For example, how do the authors know how much should an atmospheric field need to rotate/scale/move?

The individual maps were transformed to bring them to a common basis, so that in each one the cyclone center would be north of the train center, 500 Km apart. The first step was to rotate each map about the location of the cyclone's center (which served as a 'reference point') by an angle equal (and opposite) to the deviation of the orientation of the line connecting the centers to the train and the cyclone from the south-north direction. The second step was to rotate the wind direction accordingly in every grid point. The third was to stretch (or contract) the grid to bring the distance between the centers of the train and the cyclone to 500 Km. The above phrase is inserted in the revised paper.

Fig. 2: I was originally a bit puzzled by the red line as the train but then when I saw Fig. 3f, it makes perfect sense. Would the authors consider moving Fig. 3f to Fig. 2?

Thank you for your suggestion. We have done it.

Lines 156-158: The authors suggested that there is a positive relation between longitude of the cyclone center and the train orientation. However, it is not very clear to me whether this is the case. Perhaps the author could show the result of correlation analysis to support their statement?

We added the correlation, which is 0.41, confident at the 0.10 Level, to the text. For the reader convenience, the relevant columns in Table 1 appear now side by side.

Lines 160-162: I think it might be worthwhile to include some figures in the supplementary to support these statements, e.g. overlay of 925 hPa GPH and precipitation of those events.

Our finding concerning the absence of rain south of the trains is based on the radar images. We derived maps that combine the integrated radar data (as shown in Fig. 3f), with their limited areal coverage, superimposed on the 925 GPH maps, covering the Levant region,

which reflect the synoptic context. Below is the matrix of 4 maps to be included as new fig no. 6.

[Figure]

Table 1: Perhaps the event entries should be in chronological order (event no. 15 and 16).

Corrected.

Section 4:

- It would be better to separate Summary and discussion. It is because certain core ideas of this study only appear in this section (and in the abstract), which should have been discussed in earlier sections. E.g. the notion of cold trains and warm trains only appears in the abstract and in Section 4. The authors should introduce such notion in the introduction to increase readability. Furthermore, the dynamical framework of "cold train" is one of the core results of this study. This alone deserves its own section.

We agree with you, and before the last paragraph of the introduction we added: "It is worth noting that this train, and the trains analyzed in the present study, were found within the cold sector of mid-latitude cyclones, unlike the trains analyzed in North America and Europe, which were found within warm air masses. Therefore, we entitle the trains analyzed here 'cold trains', in contrast with the 'warm trains' previously studied."

Section 4 is now reorganized and merged with the Appendix. The new section 4, entitled "Dynamical considerations", contains the description of the train, as appear in the beginning of the present Sec. 4, together with the two paragraphs dealing with the

dynamics, and the content of the appendix (as you asked us to do in your last comment). The rest of present section 4 will be entitled: "5. Summary and discussion".

- [related to introduction] In the introduction, the authors should explicitly indicate the existing "warm train" model could not explain the "train" observed over EM. Then this section would make more sense.

  We reformulated the end of the last paragraph, now reads: "It is worth noting that this train, and these analyzed in the present study, were found within the cold sector of mid-latitude cyclones, unlike the trains analyzed in North America and Europe, which were found within warm air masses and can not being explained by the same mechanisms. We entitle them 'cold trains', in contrast with the 'warm trains' previously studied."

Lines 231-236: I am a bit confused. At the beginning of the paragraph, it seems that "friction exerted by the sea surface" is not a contributing factor but at the end it appears to suggest that it is?

The relevant phrase from the paper presents the problematics of friction over sea surface as follow: "The friction exerted by the sea surface seems questionable. Indeed, during calm conditions smooth sea surface produce negligible friction. But, under the influence of a CL, with its induced winds, waves are created, which may enhance friction". It stresses that just under stormy conditions, such as these associated with CLs, the rough sea exerts considerable friction. This statement relies on data of sea state during the cases analyzed (see line 234 in the original version).

Figure 6: What level(s) of pressure, wind, etc are shown in the figure?

This figure represents the lower-levels. The isobars relate to the sea level pressure and the wind field describe the lower 1 Km. For the sake of clarity, we replaced the phrase "Scheme of the train associated…" in the caption to "Schematic description of the lower-level configuration of the train associated." The phrase "the maximum wind" is now extended to "the maximum wind within the lower 1 Km".

Lines 269-274: Could this research be applied to improve nowcasting/short-range forecast?

Thank you. We add this notion to the end of the summary. "The results of our study identifies a type of regions that are prone to continuous heavy rains through the wind field. The precision of the regional models in mapping the meso-scale structure of the wind field enables us to point at the left side of maximum wind bands south of CLs as potential regions for rain inducing flash floods."

Appendix A: This section is not mentioned in the main text. Since it is related to the train formation, perhaps the authors should consider including this section in the main text rather than in the appendix.

We accept your suggestion, and included it in the new section 4, dealing with dynamics as mentioned above.

CC

General

This article presents a comprehensive investigation of rain events associated with the so called "Train Effect" phenomenon, observed in the eastern Mediterranean. The authors analyze dynamical factors and phenomenological features of 17 events, all found in the southern periphery of occluded Mediterranean cyclones, using case study analysis and composite maps. The article presents two new findings and subsequent insights, for which it deserves to be published. One, the essential differences between "East-Mediterranean trains" and those studied in the US and Western Europe. The second, the difference characterized between "coastal front" events studied in the coastal area of Israel (by Rosenfeld and Nirel 1996) and the train phenomenon studied in the presented article. The article establishes a theoretical framework for explaining "trains" as an inherent part of active cloudiness associated with occluded East Mediterranean Cyclones (i.e., Cyprus Lows). Therefore, I recommend publishing the article in Nat. Hazards and Earth System Sciences, after some weaknesses and mistakes are corrected, as specified below.

**Major Comments**

1. The study included 17 events out of 30 identified. It is essential to refer to the 13 events that were not investigated and explain why they were not investigated. It must be explained whether they represent a different phenomenon or dynamic process, which may change the insights, or whether they were less prominent in their characteristics.
   We separated the 17 cases from the other 13. We added to the first paragraph of the results (#3) sector the following: "All of them were found to delineate the southern boundary of the rainy sector within the cold sector of CLs, in the vicinity of band of maximum wind. These 17 trains were separated from the other 13 trains, which were not attributable to any other common one, and constitute the 'study sample'"

2. The authors note that based on previous studies: "the train effect is associated with quasistationary cold fronts or within pre-frontal warm tongues…" (line 39). The dynamic factors and process leading to the formation of trains associated with quasi-stationary cold fronts in the cold front must be explained, this is to distinguish it from the different dynamical factors associated with Cyprus Lows.
   The difference between the two train categories is addressed before the last paragraph of the introduction and elaborated in the first paragraph of new section 5 and Table 2 (note phrases in red).

**Specific comments:**

Page 2, lines 43-44: the authors mention "Train effect was identified also in Western Europe, near the western Mediterranean". Be more specific about the location.

Done. We added "over the Iberian Peninsula and France" in the text.

Page 2, lines 45-46: "ZAMG (2014) analyzed 100 convective systems during the period 1992 – 2009…". Specify the location of this study.

After rereading the entire paragraph, we found that this sentence does not contribute to the content, and omitted it. At the same time, we found that the 7 cases you refer to in your previous comment were taken from ZAMG (2014), and corrected it accordingly (see our previous response).

Page 2, line 55: "from the North African coast…" change to: "from the North African and North Sinai Desert coasts…"

Done.

Page 2, line 57: "…the rain produced by this type of system lasts 20 hours…" For the sake of clarity specify which type of system he refers to.

We replaced the phrase "this type of system lasts" by "this type of coastal fronts lasts".

Page 2, line 67: "…within the cold sector of a CL in 2018." Add April 2018 and stress that 'cold trains' occur not only in the mid-winter but also during spring.

"April" was added in the description of the case. In the beginning of Sec. 5, we added the following sentence: "They were found mainly in the winter, December – February, though such an event was documented in the spring (see section 1)."

Page 3, line 70: "…to document trains". I recommend changing to "… identify, document and analyze…"

The relevant phrase is now: ".. identify and document trains that are associated with CLs, and to analyze .. ".

Page 3, line 76: "…four consecutive winters (December-February) of the years 2018-2022 (except January-February 2019 due to missing radar data)." Change to: "…four consecutive mid-winters (December-February) of the years 2018-2022 (except January-February 2019 due to missing radar data), total of 10 months."

Done.

Page 3, lines 80-83: Why did the authors utilized data with high resolution, of 0.1⏢, and did not use the ERA5 data of 0.25⏢ resolution?

We had tried to use the ERA5 with 0.25° resolution, but found that it poorly reflects the signature of the trains. We added to the text the following sentence: "This data set was found superior over the ERA5 in resolving the signature of the trains in the rain field as well as in the vertical velocity field."

Page 4, line 112: "characterized by Mediterranean climate". Please specify what are the characteristics of this type of climate and add a citation.

After "..climate" we added: "type, denoted 'Cs' according to Koeppen classification, i.e., "The thermal zones of the earth according to the duration of hot, moderate and cold periods and to the impact of heat on the organic world" (Volken and Brönnimann. 2011).

Volken, E.; Brönnimann, S Meteorologische, Zeitschrift (2011). 20 (3): 351–360. Bibcode:2011MetZe..20..351K. doi:10.1127/0941-2948/2011/105", also in the reference list.

Page 4, line 112: "The rainiest area was the coastal plain, with 200-270 mm…". State how long this amount of rain accumulated and state the ratio between the amount that fell as a product of the trains (see line 120) and the total rainfall depth.

See our response to the next comment.

Page 4, line 120: "…summing up to > 60 mm for the entire event". Were there other trains that were not identified or investigated? I recommend referring to the existence of parallel trains.

Thank you for drawing our attention to the mistakes. We found out that we wrote erroneously that the train lasted for 3.5 hours, whereas it lasted only 2.5 hours. The part starting with "This system persisted 3.5 hours .." and ending with ".. exceeded 40 mm along the train" was replaced by: " This system persisted 2.5 hours and generated heavy rain, with a maximum instantaneous rate of over 60 mm h-1 (as inferred from Fig. 3a-e), summing up to >50 mm rainfall during the entire event (Fig. 2a)." Note that the rain map was moved to Fig. 2. After the sentence above, we added: "Another train occurred over the region in the previous day (Table 1, No, 1) and yielded an average rainfall of 50 mm (not shown). This implies that the two trains contributed ~35% of the total rainfall obtained over the coastal plain during the 3 days of the rainstorm."

Page 5, lines 136-137: "…was developed within a homogeneous air-mass". Recommend changing to: "…was developed within a homogeneous cold air-mass".

Done.

Page 5, line 138: "…indicate that the CL was at its occluded phase." I recommend adding a sentence (here or in the discussion section) that explains the difference from the quasistationary cold front, mentioned in the introduction section.

The low and homogeneous temperature field over the eastern Mediterranean, as seen in Fig. 2b and described in the text (see line 153). This indicates that the situation is not related to frontal activity, as appears now in lines 194 - 197: "The 850 hPa temperature (Fig. 4c) shows a homogeneous temperature over the EM, around $3°C$, which is bellow the long-term mean foe each of the months included in the study period." The comparison with the frontal trains are elaborated in the beginning of the discussion section (now 5).

Page 7, lines 161-162: "In 13 of the 17 events, no rain was observed to the right (south) of the train." And in the other four events, was there a difference in precipitation activity?

We agree with you, and shall added after that sentence: "Also, in the remaining four events, the rain activity south of the train was much weaker than to the north of it."

Page 8, Table 1: titles: add ⬜N to the Lat and ⬜E to the Lon.

Done.

Page 8, Table 1: It is recommended to add the standard deviation in addition to displaying the average only.

Standard deviations are now added. We also mention the STD of the train orientation while comparing it to that of the coastal fronts in Sec. 5. The relevant phrase is: ".. with a standard deviation of 26° (Table 1)."

Page 9, Fig. 4: After a thorough inspection I have noticed that the location of the train left of the maximum wind takes place only on the lower level (925-hPa level). Please, point to the fact that the mechanism responsible for the train originates from the below. Moreover, the figure will become clearer if you add notation of the pressure-level on each part of it. Also, why is the temperature (or temperature anomaly) plot not shown at the 850-hPa level and the 500-hPa GPH and vorticity, as shown in Figure 2?

Thank you for your valuable comment. We replaced the 925-hPa temperature field by the 850-hPa. Also, the 700 hPa wind chart was replaced by the 500-hPa wind field. The linkage between the train and the band of maximum wind, which was pronounced in the lower-levels, was not found in the upper-levels. The above is addressed in the revised manuscript: "At the medium and higher-levels, as exemplified by the 500 hPa wind field (Fig. 4a), such a linkage can not be noted. This implies that the mechanism responsible for the train formation is found within the lower-levels, see discussion below."

Page 10, Fig. 5: For the sake of clarity, I suggest adding notation of the pressure-level and atmospheric variable in parts a and b.

We agree with you, and added them in all maps presented along the paper.

Page 11, line 234: "…show an average significant height of 3-4 meters, with an average maximum of 5-6 meters". A citation is needed.

Thank you for notifying us for that. After further data elaboration the details were modified, though still the implications are similar. The pertinent phrase is now " .. significant height of 2.41±0.61 meters, with an average maximum of 3.22±0.67 meters. This implies sea state 'rough', suggesting that friction cannot be ignored. The data source is the "Coastal and Marine Engineering Research Institute Ltd (CAMERI)" in Israel (https://www.cameri-eng.com/). We shall denote and acknowledge it accordingly.

Page 12, lines 264-265: "…a small frequency compared to the 17 train events…" Shouldn't it be 30 events?

We referred to the 17 events alone, because only these were found to have affinity with the coastal fronts, and hence were candidates for comparison with the latter.

**Typographical errors**:

Page 1, line 19: change EM trains to East Mediterranean trains

Corrected.

Page 1, line 27: Saaroni et al 2011 should be corrected to Saaroni et al. 2010 and Sandler et al. 2023 should be corrected to Sandler et al.2024.

Corrected.

Page 1, line 32: delete the sign ' after the word mostly. Page 2, line 43: Change "tend"

to "tends".

Corrected.

Page 2, line 45: Change "in the form of train" to "in the form of trains".

Corrected.

Page 2, line 46: "in the form of A train" (missing "a") or "in the form of TRAINS" (in plural).

Corrected.

Page 2, line 50: Change "train effect has been…" to "train effect have been".

The word "effect" is the subject. It is single, so "has" is OK.

Page 3, line 68: Change "took lives of 10 people" to "took the lives of 10 people".

Corrected.

Page 3, line 71: Change "attempt" to "attempts".

Corrected.

Page 3, line 77: "The rain data is based" to "The rain data are based".

Corrected.

Page 3, line 87: Change "These are regarded" to "These are regarded as".

Corrected.

Page 3, line 94: Change "modified as to fit" to " modified to fit".

Corrected.

Page 3, line 95: Change "location of the CLs' centers" to "locations of the CLs' centers".

Corrected.

Page 5, line 134: Change "the cloud were stirred" to "the clouds were stirred".

Corrected.

Page 6, line 149: Change "5 of them are" to "5 of them were".

Corrected.

Page 7, line 157: Change "while these associated" to "while those associated".

Corrected.

Page 7, line 159: Change "flashflood" to "flash flood".

Corrected.

Page 11, line 228: Change "that produce this" to "that produces this".

Corrected.

Page 12, line 240, caption of Fig. 6: add the word Desert after Sinai so it will be "Sinai Desert".

Corrected.

Page 12, line 256: "… as is reflected in 925 hPa" Should be changed to: "as is reflected in the 925-hPa temperature map".

Corrected.

Page 13, line 269: Cl should be corrected to CL.

Corrected.

Page 13, line 272: Change "ingredient the CL" to "ingredient of the CL".

Corrected.

Lines 35, 40, 229: Change Chappel to Chappell.

Thanks! Corrected.

Lines 31, 38, 229: Doswell 1996 should be changed to Doswell et al. 1996.

Thanks! Corrected.

Delete the comma after the author's name in the following citations: L. 31: Doswell et al. 1996; L. 35: Chappell 1986; L. 41: Corfidi 2003; L. 77: Alpert et al. 2004; L. 80: Haiden et al. 2011; L. 83: Hólm et al. 2016; L. 92: Sinclair and Revell 2000; L. 114: IMS publication 2018.

We left the comas, because they are commonly used in "Nat Hazard".

Page 14, line 316: delete the reference of Alpert et al. (1990) since it does not appear in the text.

Deleted.

Page 14, lines 323-324: The link to Chappell (1986) cannot be found and should be: https://link.springer.com/chapter/10.1007/978-1-935704-20-1_13

Thank you! Corrected.

Page 15, line 364: The reference of Saaroni et al should be 2010 (and not 2011). Page 15, line 365: The reference of Schwartz et al. (1990) should be moved after the reference of Sandler et al.

Corrected.

Page 15, line 368: The reference of Sandler et al., should be moved before Schwartz et al. (1990), and it needs to be corrected to: Sandler, D., Saaroni, H., Ziv, B., Hochman, A., Harnik,

N. and Rostkier-Edelstein, D: A multiscale approach to statistical downscaling of daily precipitation: Israel as a test case, Int. J. Climatol., 44(1), 59-71, 2024. https://doi.org/10.1002/joc.8315

Thank you! Corrected.